# The Effects of Radioligand Therapy on Quality of Life and Sexual Function in Patients with Neuroendocrine Neoplasms

**DOI:** 10.3390/cancers15010115

**Published:** 2022-12-24

**Authors:** Pasqualino Malandrino, Rossella Mazzilli, Giulia Puliani, Sergio Di Molfetta, Gabriella Pugliese, Soraya Olana, Anna Maria Colao, Antongiulio Faggiano

**Affiliations:** 1Endocrinology Unit, Department of Clinical and Experimental Medicine, University of Catania and Garibaldi-Nesima Medical Center, 95122 Catania, Italy; 2Endocrinology Unit, Department of Clinical and Molecular Medicine, Sant’Andrea Hospital, Sapienza University of Rome, ENETS Center of Excellence, 00189 Rome, Italy; 3Department of Experimental Medicine, Sapienza University of Rome, 00185 Rome, Italy; 4Oncological Endocrinology Unit, IRCCS Regina Elena National Cancer Institute, 00128 Rome, Italy; 5Section of Internal Medicine, Endocrinology, Andrology and Metabolic Diseases, Department of Emergency and Organ Transplantation, University of Bari Aldo Moro, 70124 Bari, Italy; 6Endocrinology Unit, Department of Clinical Medicine and Surgery, Federico II University, 80131 Naples, Italy; 7UNESCO Chair on Health Education and Sustainable Development, Federico II University, 80138 Naples, Italy

**Keywords:** neuroendocrine neoplasm, peptide receptor radionuclide therapy, health-related quality of life, sexual function, radioligand therapy, questionnaire

## Abstract

**Simple Summary:**

Cancer therapies influence a patient’s health-related quality of life (HRQoL) and sexual function. Patients with neuroendocrine neoplasm (NEN) have impaired HRQoL and sexual function because of the long-lasting disease with a high tumor burden that requires multiple treatments with different mechanisms of action. Moreover, the complexity of symptoms may be due, at least to a certain extent, to the tumor secretion and release of polypeptides and biogenic amines that cause several clinical conditions including the carcinoid syndrome. Peptide receptor radionuclide therapy (PRRT) is an effective treatment in patients with NEN. Retrospective studies and randomized clinical trials have reported that patients with NEN treated with PRRT experienced an improvement in HRQoL evaluated by self-assessment questionnaires, such as the EORTC QLQ-C30 and QLQ-GINET21. The impact on sexual function has only been investigated in part through these questionnaires. This review discusses the current knowledge regarding the changes in HRQoL and sexual function in patients with NEN treated with PRRT.

**Abstract:**

Peptide receptor radionuclide therapy (PRRT), also called radioligand therapy, is an effective antitumoral treatment in patients with neuroendocrine neoplasm (NEN). It improves the patient’s health-related quality of life (HRQoL), which is evaluated by self-assessment questionnaires. The aim of this narrative review was to report the current knowledge on the changes of HRQoL and sexual function in patients with NEN treated with PRRT. We conducted a literature search of the PubMed, Embase, and APA PsycInfo databases. We selected 15 studies (12 for HRQoL and three for sexual function). After treatment with PRRT, patients with NEN experienced a significant improvement in their global health status, disease-related worries, social and emotional functioning, and cancer-related symptoms such as fatigue and diarrhea. Other symptoms, such as nausea/vomiting, dyspnea, and constipation, as well as the economic impact, were unchanged by radioligand therapy. Data on sexual function were not equally promising; only a few studies investigated this issue by using appropriate questionnaires in patients treated with radioligand therapy. Therefore, additional studies are needed to draw a conclusion about the benefits from PRRT on sexual function.

## 1. Introduction

Neuroendocrine neoplasms (NENs) represent a heterogeneous group of neoplasms, with primary tumors commonly found in the pancreas, small intestine, and lungs [1,2,3]. Among several therapeutic options, the systemic administration of radiolabeled synthetic somatostatin analogs, namely, radioligand therapy, also referred to as peptide-receptor radionuclide therapy (PRRT), is one of the most effective and safest treatment options for well-differentiated, unresectable/metastatic NEN with an increased uptake on somatostatin receptor imaging [4]. The efficacy of PRRT for the treatment of NEN was documented by the Neuroendocrine Tumors Therapy (NETTER-1) trial. In this phase 3, multicenter, randomized controlled study, the investigators documented longer progression-free survival (PFS) in patients with World Health Organization (WHO) grade 1 or 2 midgut well-differentiated NEN, so-called neuroendocrine tumors (NETs), who were treated with 177Lu-dotatate plus octreotide long-acting repeatable than in patients receiving high-dose octreotide long-acting repeatable alone [5]. Similar results were achieved in other retrospective series and small prospective phase 2 trials [6,7,8,9]. Clinical safety data suggest that PRRT with either 90Y- or 177Lu-peptides is generally well tolerated. Indeed, acute adverse effects (nausea and vomiting) are usually mild and related to the coadministration of amino acids used to reduce kidney radiation exposure. Fatigue and exacerbation of a hormonal crisis may also occur and are consistent with the cytotoxic mechanism of the radiopeptides. Given the high prevalence of subjects who experience a heavy symptom burden at a relatively young age, with relevant effects on family life, personal finances, and the ability to work, health-related quality of life (HRQoL) is reported to be poor in patients with NEN [10]. Moreover, sexual health is strictly related to HRQoL. In fact, as underlined by WHO, it is fundamental to the overall health and wellbeing of individuals, couples, and families. Among the predictors of sexual dysfunction in patients with advanced cancer, the site of the tumor (mainly gynecological tumors), recent hospitalization, and absence of a sexual partner have been described [11]. Both advanced cancer and oncological treatments may influence HRQoL and sexual function. For example, acute and chronic adverse effects from therapies may negatively influence a patient’s HRQoL, but may also warrant an improvement secondary to tumor and hormonal response. The aim of this review was to investigate the impact of PRRT on HRQoL and sexual function in patients with NEN.

## 2. Literature Search Strategy

### 2.1. Article Identification

We searched the PubMed, Embase, and APA PsycInfo databases for studies reporting assessment of HRQoL and sexual function in patients with NEN who had undergone PRRT. We used the following search terms: “quality of life and neuroendocrine”; “peptide receptor radionuclide therapy and quality of life”; “PRRT and quality of life”; “radioligand therapy and quality of life”; RLT and quality of life”; “177 lutetium dotatate and quality of life”; “177 lutetium dotatate and sexual function”; “177 lutetium dotatate and sexual dysfuntion”; “90 ittrium and quality of life”; “90 ittrium and sexual function”; “90 ittrium and sexual dysfuntion”; “sexual function and neuroendocrine”; “sexual dysfunction and neuroendocrine”; “erectile function and neuroendocrine”; “erectile dysfunction and neuroendocrine”. Additional details on the combinations of the MeSH terms and keywords are provided as Appendix A. The last search date was 30 November 2021. We included English-language studies in humans, without a time restriction and with any of the following designs: randomized clinical trials, prospective nonrandomized trials, and retrospective studies. We screened review articles to find additional original articles on the topic. The inclusion criteria were (1) articles reporting data on HRQoL or sexual function, (2) patients affected by NEN, (3) patients who had undergone PRRT, (4) use of EORTC QLC-C30 and/or QLQ-GINET21 to assess HRQoL, and (5) studies reporting complete data for all the questionnaire items. The exclusion criteria were (1) case reports and case series (defined as less than five patients), (2) articles not in English, and (3) articles evaluating HRQoL with questionnaires other than EORTC QLC-C30 or QLQ-GINET21 (no restriction on questionnaire type for sexual function evaluation).

### 2.2. Article Selection

Six authors (S.D.M., P.M., R.M., S.P., Gi.Pu., and Ga.Pu.) screened the retrieved articles by abstract and title. They then assessed potentially eligible studies by retrieving full-length articles and evaluating them according to the inclusion and exclusion criteria. Two authors (P.M. and R.M.) extracted data from the articles that met the inclusion criteria. They used a standardized form to extract the following information: first author, study design, year of publication, number of patients enrolled, age at diagnosis, tumor site, staging and grading, possible tumor-associated secretion, type of treatment(s) used, and HRQoL and sexual function assessment (including the type of questionnaires used and timing of administration relative to PRRT). From the original 3814 articles identified, we selected 92 on the basis of the title and abstract. After full-text examination, we included 15 articles on HRQoL and/or sexual function in the review (Figure 1).

### 2.3. Questionnaires to Assess HRQoL: QLQ-C30 and QLQ-GINET21

The EORTC Core QoL questionnaire (QLQ-C30) is a widely used generic questionnaire for cancers, and it has been developed to be modular, with the core general questionnaire that can be implemented with disease-specific modules containing those aspects of HRQoL relevant to patients with a specific type or site of cancer [12]. The QLQ-GINET21 was introduced and validated recently to be used in conjunction with the EORTC QLQ-C30, with the aim of evaluating specific and relevant HRQoL issues in patients with NEN [13]. Although the EORTC QLQ-C30 is the most widely used in oncology, it does not cover specific disease- and treatment-related issues such as flushing, abdominal pain, discomfort in joints and muscles, the effect of the disease on sexual behavior, and some specific psychological, emotional, and social aspects. The Social Functioning (SF) scale of the QLQ-GINET21 is a new type of scale, in which items are ordered to evaluate how difficult SF is for the subject. The SF scale of the QLQ-C30 correlated quite highly with the SF scale in the QLQ-GINET21, suggesting that both measure and evaluate similar aspects of the same problem [14]. However, the SF scale of the QLQ-GINET21 seems to be different and, thus, provides additional information about the changes in SF that occur over time. Moreover, none of the other scales of the QLQ-GINET21 and QLQ-C30 showed a high correlation, thus confirming that the QLQ-GINET21 provides additional information beyond the more generic QLQ-C30 [14].

Both EORTC QLQ-C30 and QLQ-GINET21 are used for research in many countries and can also be applied in clinical practice, because the first has been translated into more than 100 languages and the second has been translated into nine languages. The data comparing cancers of different origins suggests that the QLQ-GINET21 can be used for both pancreatic and nonpancreatic NETs, although it is accepted that there are not enough cases of pancreatic NETs for a proper validation in patients with this cancer type alone.

Additional details on the questions included in the EORTC QLQ-C30 and QLQ-GINET21 questionnaires and how scores are calculated according to patients’ answers are provided in the Appendix A.

## 3. Results

### 3.1. Effect of PRRT on HRQoL

Twelve studies evaluated HRQoL in patients with NEN treated with PRRT (Table 1).

Eight of these studies were retrospective analyses [15,16,17,18,19,20,21,22], and four studies were prospective analyses [23,24,25,26]. ^177^Lutetium was the radioisotope administered in 11 of the 12 studies, and only in one study patients were treated with ^225^Ac-DOTATATE [24]. The cumulative administered activity of ^177^Lu-DOTATATE was quite similar among the selected studies, which ranged on the average from 22.2 to 29.6 GBq. This was achieved by administering an intended dose of 7.4 GBq per cycle up to 3–4 cycles. Larger doses were administered in the study by Del Prete et al. [26] because injected activity was calculated through a quantitative SPECT/CT-based dosimetry; therefore, the median cumulative administered activity was 36.1 GBq (range 6.3–78.6). Premedication with intravenous amino-acid infusion to protect the renal function was described in nine studies. This was performed with a similar protocol through a solution of lysine and arginine infused over 4–6 h and starting 30 to 60 min before the therapy. By considering all the selected studies, HRQoL was evaluated in a total of 1005 patients, including 524 males and 481 females, with a mean age ranging from 52 to 70 years old. The study populations included mainly patients with midgut and pancreatic NET and less frequently bronchial NET. Changes in HRQoL were analyzed only by the EORTC QLQ-C30 in 10 of the studies, only by the QLQ-GINET21 in two of the studies, and by both questionnaires in one study. Scores at baseline, before PRRT treatment, were compared with those recorded at the end of treatment (i.e., from 6 weeks to 3 months after the last PRRT cycle). The quantification of the mean changings for each item of the questionnaires was described in eight of the 12 studies, whereas these data were not available in three studies [16,21,25]; in the study by Strosberg et al. [23], the authors considered as outcome the time to deterioration, i.e., the time from random assignment to the first deterioration of 10 or more points (on a 100-point scale) compared with baseline score for the same domain. Therefore, to summarize our findings by including all the selected studies we used a graphical method. Table 2 and Table 3 display the changes from the baseline values of each item score of the QLQ-C30 and QLQ-GINET21, respectively.

**Table 1 cancers-15-00115-t001:** Main features of selected studies on quality of life in patients with neuroendocrine tumors treated by peptide receptor radionuclide therapy.

	Teunissen et al. (2004) [15]	Khan et al. (2011) [16]	Bodei et al (2011) [25]	Marinova et al. (2018) [17]	Martini et al. (2018) [18]	Strosberg et al. (2018) [23]		Del Prete et al. (2018) [26]	Marinova et al. (2019) [19]	Zandee et al. (2019) [20]	Ballal et al. (2020) [24]	Chen et al. (2021) [21]	Zandee et al. (2021) [22]
Study design	Retrospective analysis	Retrospective analysis	Prospective analysis	Retrospective analysis	Retrospective analysis	Randomized phase 3 study		Prospective analysis	Retrospective analysis	Retrospective analysis	Prospective analysis	Retrospective analysis	Retrospective analysis
Intervention	^177^Lu-DOTATATE	^177^Lu-DOTATATE	^177^Lu-DOTATATE	^177^Lu-DOTATATE	^177^Lu-DOTATATE (67%) or ^90^Y-DOTATOC (33%)	^177^Lu-DOTATATE	Octreotide LAR 60 mg every 4 weeks	^177^Lu-DOTATATE	^177^Lu-DOTATATE	^177^Lu-DOTATATE	^225^Ac-DOTATATE	^177^Lu-DOTATATE	^177^Lu-DOTATATE
Cumulative administered activity (GBq)	22.2–29.6	22.2–29.6	25.2 (median)	28.2 (median)	Not reported	29.6	-	36.1 (median)	27.4 (median)	29.6 (median)	55.5 kBq	27.5 (mean)	26.8 (median)
Intravenous amino acid premedication for renal protection	Not reported	Not reported	25 g of lysine diluted in 1 L of normal saline infused over 4 h, starting 30–60 min before, followed by an additional 12.5 g of lysine diluted in 500 mL of normal saline, over 3 h, twice a day on day 2 and day 3 post therapy	2.5% lysine and 2.5% arginine in 1 L of 0.9% NaCl, for over 4 h, starting 30 minutes before therapy	Not reported	21.0 g of lysine and 20.4 g of arginine in 2 L of solution or 18 g of lysine and 22.6 g of arginine in 2 L of solution, for at least 4 h, starting 30 minutes before therapy	-	25 g of L-lysine dihydrochloride and 25 g of L-arginine dihydrochloride dis- solved in 1 L of normal saline, infused over 4 h	2.5% lysine and 2.5% arginine in 1 L of 0.9% NaCl, for over 4–6 h, starting 30 min before therapy	2.5% lysine and 2.5% arginine in 1 L of 0.9% NaCl, for over 4 h, starting 30 min before therapy	Solution of lysine and arginine infused over 4 h, starting 30 to 60 min before the therapy	2.5% lysine and 2.5% arginine in 1 L of 0.9% NaCl, for over 4–6 h, starting 30 min before therapy	2.5% lysine and 2.5% arginine in 1 L of 0.9% NaCl, for over 4 h, starting 30 min before therapy
No. of patients	50	265	51	68	61	116	113	52	70	34	32	71	22
Gender (M/F)	22/28	137/128	26/25	37/31	37/24	63/53	53/60	24/28	39/31	17/17	15/17	42/29	12/10
Mean age (range)	58 (30–78)	59 (23–83)	56 (30–79)	61 (14–85)	62 (37–88)	63	64	55 (17–78)	64 (34–83)	59	52 (35–72)	70 (55–80)	63
Primary site	GEP (86%), unknown origin (14%)	GEP (36%), bronchial NET (64%)	GEP (76%), bronchial (10%), others (8%), un-known origin (6%)	Pancreas (100%)	GEP (100%)	Midgut (100%)		GEP (77%), bronchial (6%), others (9%), un-known origin (8%)	Midgut (100%)	Pancreas (100%)	GEP (84%), unknown origin (16%)	GEP (89%), bronchial (4%), unknown origin (7%)	Midgut (100%)
Functioning NET	Gastrinoma (6%), insulinoma (2%)	4%	55%	32%	Not reported	Not reported		63%	83%	100%	Not reported	59%	100%
Prior therapy	Surgery (44%), chemotherapy (10%), SSA (48%)	Surgery (18%), chemotherapy (9%), SSA (62%), radiotherapy (5%)	Surgery (68.6%), chemotherapy (21.6%), SSA (84.3%)	Surgery (54%), chemotherapy (38%), SSA (37%), locoregional (10%)	Surgery (54%), chemotherapy (18%), SSA (72%), locoregional (8%), other (16%)	Surgery (80%), chemotherapy (9%), SSA (100%), locoregional (19%), other (30%)	Surgery (82%), chemotherapy (12%), SSA (100%), locoregional (17%), other (26%)	Surgery (56%), chemotherapy (29%), SSA (77%), locoregional (25%), other (40%)	Surgery (71%), chemotherapy (11%), SSA (73%), locoregional (7%)	Surgery (29%), chemotherapy (9%), SSA (65%), other (21%)	Surgery (31%), chemotherapy (38%), SSA (88%), ^177^Lu-DOTATATE (100%)	Chemotherapy (14%), SSA (93%), other (4%)	Surgery (59%), chemotherapy (9%), SSA (100%), locoregional (14%), other (23%)
Progressive disease before PRRT	34%	48%	76%	72%	30%	100%	100%	100%	79%	65%	56%	80%	0%
Tumor response to PRRT	CR or PR (48%), SD (38%), PD (12%)	CR or PR (45%), SD (35%), PD (14%)	CR or PR (29%), SD (53%), PD (18%)	Not reported	Not reported	CR or PR (18%)	CR or PR (3%)	PR (59%), SD (33%), PD (8%)	Not reported	CR or PR (58%), SD (24%), PD (18%)	PR (63%), SD (37%)	PR (15%), SD (79%), PD (6%)	PR (9%), SD (68%), PD (23%)
HRQoL questionnaire	EORTC QLQ-C30	EORTC QLQ-C30	EORTC QLQ-C30	EORTC QLQ-C30	EORTC QLQ-C30	EORTC QLQ-C30 and QLQ-GINET21		EORTC QLQ-C30	EORTC QLQ-C30	EORTC QLQ-C30	EORTC QLQ-GINET21	EORTC QLQ-GINET21	EORTC QLQ-C30
Timing of HRQoL assessment for statistical analysis	The week before the first treatment and 6 weeks after last treatment	Baseline and 6 weeks after last treatment	Baseline, at treatment cycles and at 3, 12, and 24 months of follow-up	Baseline and 3 months after the last (fourth) PRRT cycle.	Baseline and 3 months after the last PRRT cycle.	Time from random assignment to first deterioration of 10 points in domain score compared with baseline score for the same domain		Baseline and at 3 months after the last PRRT cycle	Baseline and 3 months after the third PRRT cycle	Baseline and 3 months after the last (fourth) PRRT cycle	Baseline and at the time of analysis	Baseline and after each PRRT cycle	Baseline and 3 months after the last (fourth) PRRT cycle

According to the findings from the QLQ-C30 (Table 2), nine of the 10 selected articles showed a significant improvement in the global health status. For instance, Zandee et al. [20] reported that, after PRRT, the global health status score increased by 17.8 points from the baseline. The only study reporting a neutral effect of PRRT on the global health status [22] was conducted on 22 patients, and the results of the QLQ-C30 were available for only 12 patients. This small sample may explain why the authors did not observe any benefit from PRRT. Among the items investigating the functional scales, we observed that, in more than half of the studies, the social and emotional functioning was ameliorated at the end of the PRRT. This improvement was higher for the social functioning (the mean score increased by 7.0 to 37.1 points from the baseline) than for the emotional functioning (the mean score increased by 8.1 to 17.1 points from the baseline). Similarly, there was a certain improvement for fatigue and diarrhea. In this case, the improvement was more evident for the fatigue mean score (−9.1 to −27.7 points from the baseline) than for the diarrhea mean score ( −9–9 to −16.3 points from the baseline). Relative to the baseline scores, the following items were unchanged after PRRT in all the selected studies: nausea/vomiting, dyspnea, constipation, and economic impact. As shown in Table 3, three studies applied the EORTC QLQ-GINET21 to analyze HRQoL [21,23,24]. In all these studies, the authors found an improvement in the disease-related worries in patients treated with PRRT. For the other items, the findings were quite inconsistent. For example, Strosberg et al. [23] observed some benefit for body image, whereas Ballal et al. [24] reported a lower score of the body image scale after PRRT. Moreover, endocrine and gastrointestinal symptoms were not influenced by PRRT in the studies by Strosberg and Chen [21,23], whereas Ballal et al. [24] documented a significant improvement with respect to the baseline scores.

### 3.2. Effect of PRRT on Sexual Function

The QLQ-GINET21 includes one question about sexual function: “Has the disease or treatment affected your sex life (for the worse)?” On the basis of this question, Ballal et al. [24] showed an improvement in sexual function in 38 patients with gastroenteropancreatic NET who had undergone PRRT, although it was not significant (*p* = 0.678). Similarly, Strosberg et al. [23] found no significant differences in time to HRQoL deterioration, defined as the time from random assignment to the first HRQoL deterioration ≥10 points for each patient, in the sexual function domain, when comparing PRRT with somatostatin analogs (score 30.6 vs. 28.2; *p* = 0.7). Karppinen et al. [27] explored sexual function in patients with small intestine NEN; of them, 26.9% were treated with PRRT. The diagnosis was made through the 15D questionnaire [28], which explores HRQoL with only one specific question on sexual activity. Specifically, this instrument comprises 15 dimensions: mobility, vision, hearing, breathing, sleeping, eating, speech, excretion (includes both bladder and bowel function), usual activities, mental function, discomfort and symptoms, depression, distress, vitality, and sexual activity. The authors found that PRRT did not predict or correlate with impaired HRQoL or sexual function. Furthermore, van der Horst-Schrivers et al. [29] found that patients with well-differentiated NEN did not experience sexual problems more often than a reference population. However, the sample was limited to patients with metastatic midgut NEN, and the questionnaire used was the same for both male and female patients, namely, the short version of the Questionnaire for Screening Sexual Dysfunction (QSD) [30]. This instrument assesses the frequency and experienced distress of sexual problems on different subscales, such as orgasm, erection, arousal, and pain. Interestingly, plasma tryptophan levels were lower and urinary 5-HIAA concentrations were higher in patients with sexual dysfunction. The authors concluded that these patients represent the population with more extensive and longstanding disease and, therefore, are more affected by sexual dysfunction [29]. However, no patient was treated with PRRT. Lastly, limited data are available regarding sexual dysfunction in female patients with neuroendocrine carcinoma of the cervix. Specifically, Zaid et al. [31] enrolled a heterogeneous group of patients through social media from eight countries across four continents and reported similar prevalence of sexual dysfunctions in the study group compared with the controls. However, only 50% of the patients had filled out the items regarding sexuality, explored by the Patient-Reported Outcomes Measurement Information System (PROMIS) on sexual functioning [32]. Of note, the authors did not perform a sub-analysis on the population that had received PRRT.

Although sexual health is an integral part QoL, both in the general population and in the cancer population, this aspect has not been sufficiently investigated in NET patients. Moreover, to date, no data are available regarding the study of sexual function through the use of more specific questionnaires (i.e., International Index of Erectile Function (IIEF)-5 or -15 questionnaires, as well as the specific item in the Beck Depression Inventory (BDI)-II questionnaire).

## 4. Discussion

Cancer therapies are aimed at controlling tumor growth and cancer-related symptoms. Moreover, NENs are often characterized by hormone production and release that eventually influence the patient’s symptoms. Therefore, available cancer treatments for patients with NET should be addressed to control three main targets: tumor growth, hormone secretion, and the patient’s symptoms. The best hypothetical therapeutic agent should not negatively influence an additional fourth parameter that is closely related to the abovementioned three parameters, i.e., the patient’s quality of life and sexual function.

In recent years, PRRT has dramatically changed the management of patients with advanced NET by offering them a safe and effective therapeutic option. In this review, we focused our efforts to determine whether the benefits of PRRT also extend to HRQoL and sexual function. We identified a number of manuscripts with sufficient data for clinical consideration of this matter. Most of the selected studies were consistent with a general improvement in HRQoL. This finding concerns the global health status and some scales regarding emotional and social functions. Interestingly, the global health status has been considered a predictor of a good prognosis in patients with advanced cancer [33,34]. Teunissen et al. [15] reported better quality of life in patients who had achieved tumor regression after PRRT. Similarly, as shown in Table 2 and Table 3, the scores for several items concerning the functional and symptom scales were improved in series in which patients had had a high percentage of complete or partial response [16,20,24]. Additionally, in about half of the selected studies, PRRT was associated with a significant reduction from baseline in symptom seriousness such as fatigue and diarrhea. The reduction in diarrhea after PRRT was obtained despite previous treatment with somatostatin analogs in most patients. Noteworthily, all studies also documented a neutral effect on nausea/vomiting, dyspnea, and constipation. This finding may have a double interpretation. Indeed, on the one hand, we can speculate on the good toxicity profile of the PRRT; it is generally well tolerated and only rarely worsens the patient’s symptoms. On the other hand, we should expect a certain improvement after the treatment, but this was not observed because the abovementioned symptoms are in most of cases due to an obstructive syndrome, e.g., secondary to the typical mesenteric desmoplastic reaction, which may be not responsive to PRRT. Moreover, the EORTC QLQ-C30 was developed to assess HRQoL in patients with cancer regardless of the primary site [12] and may not be totally appropriate to investigate all issues concerning HRQoL in patients with NEN. Therefore, the EORTC QLQ-GINET21 was developed in a phase 1–3 study to supplement the QLQ-C30 and to provide a more specific assessment of disease and treatment-related issues in patients with NEN [13]. By using the GINET21 questionnaire, Ballal et al. [24] documented that endocrine and gastrointestinal symptoms had been ameliorated after PRRT, whereas, in the phase 3 NETTER-1 trial, these symptoms were not different from the baseline after PRRT [23]. Recently, a secondary analysis of the NETTER-1 trial reported that PRRT was associated with a significant reduction in diarrhea, flushing, and abdominal pain [35]. For this analysis, the authors investigated symptoms that patients had recorded in a daily diary, namely, general symptoms, respiratory symptoms, and gastrointestinal symptoms. This apparent inconsistency may be explained in part by the different conception and aim between the daily dairy, where the patient reports their experience day by day, and the EORTC questionnaire, with summarizes the patient’s symptoms occurring over a longer time (i.e., 1–4 weeks). At the 2020 ENETS Annual Conference, researchers presented the phase 1–3 study for developing a new questionnaire specific for patients with pancreatic NEN (panNEN) [36]. According to the literature review and healthcare experts, researchers from seven countries developed a provisional phase 3 questionnaire and tested it on 59 patients with functioning (gastrinoma and insulinoma) or nonfunctioning panNEN [37]. Because panNENs display some biological and clinical differences from the other gastrointestinal NENs, the QLQ-C30 does not address the issues deriving from functioning NENs, and the QLQ-GINET21 is not dedicated to panNEN, this new questionnaire, when validated, will be helpful to fill this lack in the next future.

In a recent review and meta-analysis, the authors investigated HRQoL in patients with NEN treated with different therapies, including somatostatin analogs, PRRT, chemotherapy, and targeted therapy [38]. The authors reported that HRQoL was stable in patients who had been treated with somatostatin analogs alone or chemotherapy, whereas it improved in patients who had been treated with PRRT or targeted therapy. This observation is consistent with the findings of this review; therefore, the therapeutic sequence in patients with NEN should be driven not only by expected antitumor activities, but also by the effect on HRQoL changes. Moreover, body image is an important aspect of HRQoL of cancer survivors [39], and, in this regard, PRRT could be a treatment with a low impact on physical appearance and a good perceived global health status.

The long natural history of NENs, as well as the necessity of long-lasting and stepwise therapies, could have a negative impact on the patient’s sexual health. These events may occur independently of the primary site and patient’s sex. In fact, the cancer and its treatments have consequences for the sexuality of the patients and their partners, and it could be associated with detrimental effects on erection, ejaculation, and orgasm in male, as well as decreased desire in female [40]. However, studies exploring sexual functioning, sexual health, and sexual satisfaction in patients with NEN are limited [24,27,29,31]. Moreover, data on this issue in patients treated with PRRT are derived from a single question included in the QLQ-GINET21 that investigates whether the disease or treatment has affected the patient’s sex life during the past 4 weeks. The only two studies with available data [23,24] reported that PRRT did not influence sexual function. Interestingly, a recent study highlighted that carcinoid syndrome (CS) could negatively affect sexual function [41]. It is known that 5-hydroxy-tryptamine (5-HT), which is recognized as an important modulator of female and male sexual ejaculatory/orgasmic function, is increased in CS, and a higher level of 5-HT inhibits erectile function, lubrification, and sexual interest [42]. Furthermore, male subjects affected by CS had a higher serum luteinizing hormone concentration, as well as a decreased libido and erectile dysfunction, even with normal testosterone concentrations [43]. However, according to the results of this review, additional studies considering sexual dysfunctions in patients with NEN are needed.

This review had some limitations mainly due to the heterogeneity of the HRQoL evaluation among the selected studies (see the last row of Table 1). However, to minimize this measurement bias, in each study, we compared only the pretreatment HRQoL estimation with the values obtained after the last PRRT cycle. Furthermore, we expected a different HRQoL response to PRRT relative to the different percentage in each series of patients with functioning NEN. We speculate that the impact of this element should not be relevant because, in all studies, most patients were on somatostatin analog therapy before they began PRRT. Lastly, we could have selected more studies by considering other questionnaires used in patients with NEN (i.e., the Norfolk QoL-NET), but we only focused on the two most used questionnaire with the aim of obtaining more homogeneous data.

## 5. Conclusions

In conclusion, HRQoL assessment may help clinicians to perceive a patient’s needs, providing a guide to select the most appropriate treatment for each patient. PRRT is not only effective and safe, but may also allow an improvement in HRQoL in patients with NEN. However, more accurate and personalized evaluations, especially on sexual function, are still needed; therefore, a new HRQoL questionnaire should be validated for this purpose.

## Figures and Tables

**Figure 1 cancers-15-00115-f001:**
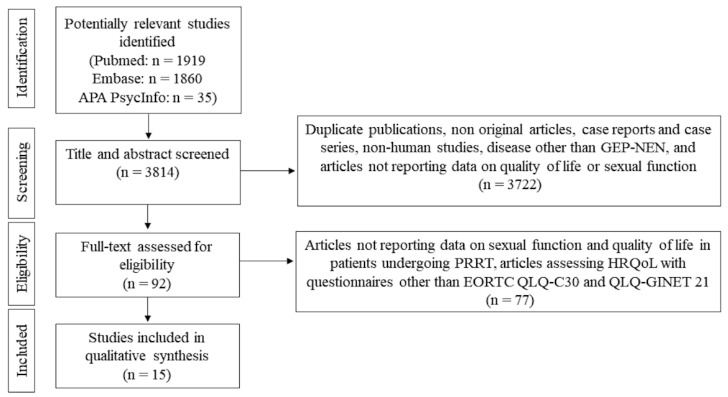
Flowchart showing the study selection.

**Table 2 cancers-15-00115-t002:** Changes of quality of life in patients with neuroendocrine tumors treated with PRRT: comparison with pre-therapy EORTC QLQ-C30 scores.

	Teunissen et al.(2004) [15]	Khan et al.(2011) [16]	Bodei et al.(2011) [25]	Marinova et al.(2018) [17]	Martini et al.(2018) * [18]	Strosberg et al.(2018) [23]	Del Prete et al.(2018) [26]	Marinova et al.(2019) [19]	Zandee et al.(2019) [20]	Zandee et al.(2021) [22]
**EORTC QLQ-C30 Items**					**SI-NET**	**P-NET**					
**Global health status**					**~**	 **					**~**
**Functional scales**											
Physical	**~**	**~**	**~**	**~**		 **		**~**	**~**		**~**
Role		**~**	**~**	**~**	**~**	**~**		**~**	**~**		**~**
Emotional			**~**	**~**	**~**		**~**	**~**			**~**
Cognitive	**~**	**~**	**~**	**~**	**~**	 **	**~**	**~**	**~**	**~**	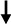
Social			**~**				**~**		**~**		**~**
**Symptoms scales**								**~**			
Fatigue		**~**			**~**	 **		**~**	**~**		**~**
Nausea/vomiting	**~**	**~**	**~**	**~**	**~**	**~**	**~**	**~**	**~**	**~**	**~**
Pain		**~**	**~**	**~**	**~**	**~**		**~**	**~**	**~**	**~**
**Single items**											
Dyspnea	**~**	**~**	**~**	**~**	**~**	**~**	**~**	**~**	**~**	**~**	**~**
Insomnia			**~**	**~**	**~**	**~**	**~**	**~**		**~**	**~**
Appetite loss	**~**					**~**	**~**	**~**	**~**	**~**	**~**
Constipation	**~**	**~**	**~**	**~**	**~**	**~**	**~**	**~**	**~**	**~**	**~**
Diarrhea	**~**			**~**		**~**		**~**		**~**	**~**
Economic impact	**~**	**~**	**~**	**~**	**~**	**~**	**~**	**~**	**~**	**~**	**~**


 = significant improvement of the item score compared with the baseline; **~** = no difference of the item score compared with the baseline; 
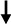
 = significant worsening of the item score compared with the baseline. * QLQ-C30 scores were compared with age- and sex-matched population-based controls. ** These changes were observed only in patients treated by ^177^Lu. SI-NET= small intestine neuroendocrine tumor; P-NET= pancreatic neuroendocrine tumor.

**Table 3 cancers-15-00115-t003:** Changes in quality of life in patients with neuroendocrine tumors treated by PRRT: comparison with pre-therapy EORTC QLQ-GINET21 scores.

EORTC QLQ-GINET21 Items	Strosberg et al. (2018) [23]	Ballal et al. (2020) [24]	Chen et al. (2021) [21]
Endocrine symptoms	**~**		**~**
Gastrointestinal symptoms	**~**		**~**
Treatment-related symptoms	**~**	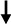	**~**
Social function	**~**	**~**	**~**
Disease-related worries			
Muscle and/or bone pain	**~**	**~**	Not reported
Body image		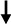	Not reported
Information	**~**	Not applicable	Not reported
Sexual function	**~**	**~**	Not reported
Weight gain		**~**	Not reported


 = significant improvement of the item score compared with the baseline; **~** = no difference of the item score compared with the baseline; 
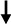
 = significant worsening of the item score compared with the baseline.

## Data Availability

The data presented in this study are available in the article and tables.

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
