# Peer review of "The Effects of Radioligand Therapy on Quality of Life and Sexual Function in Patients with Neuroendocrine Neoplasms"

_cancers, 2022, doi:10.3390/cancers15010115_

Round 1
Reviewer 1 Report (Previous Reviewer 1)
I find the authors have adequately answered reviewers comments.
Reviewer 2 Report (Previous Reviewer 2)
Thank you for the much improved revision. No further comments.
This manuscript is a resubmission of an earlier submission. The following is a list of the peer review reports and author responses from that submission.
Round 1
Reviewer 1 Report
In general, I think it is a good, very straightforward review of the current knowledge of Quality-of-Life effects of DOTATATE radionuclide therapy. One could question whether a review is necessary with so few relevant papers, but it could be a good summary if nothing else.
Radionuclide therapy is a treatment that can be given in different amounts, and I am a little concerned that this paper seems to approach it as just a binary treated and untreated. There is some information about what timepoint the questionnaire is given but none if the injected activity so far to the patients have been at least similar. Do all cited studies employ the same administration of amino acids to protect the kidney for example? The pooling of treatments using different radionuclides is also a bit concerning, you would expect different responses on both tumors and risk organs depending on if you use a beta or alpha emitter and what pathlength the beta. I feel at least some discussion should be made about this, maybe to compare if the general conclusions would change if only 177Lu-studies were included.
Since the questionnaires have quantified responses, it appears strange that the Table in this review only shows if a change is significant and direction but does not speak at all what size the change is. If that information is not available in all or some of the reviewed papers then that should be spelled out in the paper. We don’t necessarily need quantification of all changes but at least some discussions about if some changes are larger than others, if an effect is small but significant due to large dataset or similar.
I would like some further discussion about why sexual function is relevant to measure. Are gynaecological tumors common, are males affected at all etc?
Author Response
Point 1: Radionuclide therapy is a treatment that can be given in different amounts, and I am a little concerned that this paper seems to approach it as just a binary treated and untreated. There is some information about what timepoint the questionnaire is given but none if the injected activity so far to the patients have been at least similar. Do all cited studies employ the same administration of amino acids to protect the kidney for example? The pooling of treatments using different radionuclides is also a bit concerning, you would expect different responses on both tumors and risk organs depending on if you use a beta or alpha emitter and what pathlength the beta. I feel at least some discussion should be made about this, maybe to compare if the general conclusions would change if only 177Lu-studies were included.
Response 1: We thank the reviewer for this good suggestion. We modified Table 1 to specify the cumulative injected activity and the amino acid premedication protocol for renal protection. The text was changed accordingly (page 8, lines 154-612). 177Lutetium was the radioisotope administered in eleven of the twelve studies and only in one study patients were treated with 225Ac-DOTATATE. Manuscript has been modified to specify this (page 8, lines 152.153).
Point 2: Since the questionnaires have quantified responses, it appears strange that the Table in this review only shows if a change is significant and direction but does not speak at all what size the change is. If that information is not available in all or some of the reviewed papers then that should be spelled out in the paper. We don’t necessarily need quantification of all changes but at least some discussions about if some changes are larger than others, if an effect is small but significant due to large dataset or similar.
Response 2: We agree with the reviewer and changed the manuscript to specify if some changes were larger than other. Unfortunately, not all the studies had quantified responses. However, we calculated the changings of the mean scores when available and added some comments as suggested (page 10, lines 190-195).
Point 3: I would like some further discussion about why sexual function is relevant to measure. Are gynaecological tumors common, are males affected at all etc?
Response 3: This is a valid and important issue, and we are curious what the results would be. However, studies on this are lacking, especially for patients with NENs. Gynaecological tumors are uncommon and according to our literature search we identified a total of 1005 patients, with a slight not significant involvement of males (n.= 524) than females (n.= 481) patients. It’s our opinion that, in patients with cancer, sexual function is relevant to measure because it influences the human wellbeing. The cancer and its treatments have consequences on the sexuality of the patients and their partners. These events may occur independently of the primary site and patient’s sex. This aspect is discussed at page 11 lines 239-244 and page 10 lines 311-314.
Reviewer 2 Report
An interesting manuscript.
Few comments.
1. The impact on sexual function is very scarcely studied as the authors do not use more specific questionnaires; e.g. question no 21 in BDI-II or IIEF-5. Maybe it would be appropriate just do describe QoL and not sexual function?
Author Response
Point 1: The impact on sexual function is very scarcely studied as the authors do not use more specific questionnaires; e.g. question no 21 in BDI-II or IIEF-5. Maybe it would be appropriate just do describe QoL and not sexual function?
Response 1: We thank the reviewer for this excellent point and as suggested we have searched in the selected literature databases for published articles on QoL (assessed by both BDI-II and IIEF-5) and PRRT in patients with NET. Unfortunately, this search gave us no results. In the revised manuscript we left the results on sexual function just because we aimed at underlining that this issue has not enough attention by the current literature. Since the cancer and its treatments have consequences on the sexuality of the patients and their partners, it’s our opinion that sexual function is relevant to evaluate because it influences the human wellbeing. This aspect is discussed at page 11 lines 239-244 and page 10 lines 311-314.
Reviewer 3 Report
This is a literature review of quality of life and sexual function in patients with neuroendocrine neoplasm following treatment with peptide receptor radionuclide therapy. A significant amount of work has been done to put together this manuscript. Sexual function/dysfunction in particular is not given enough attention.
The manuscript is well-written. The presentation of the results is clear and appears accurate. The discussion appropriately summarises the literature
Unfortunately the literature search may have missed some relevant literature through the choice of database and search terms.
1. Databases such as PsychInfo and CINAHL should have been searched, especially for quality of life, sexual function literature that may not be included in the medical/scientific literature
2. abbreviations appear to have been used for search terms, eg. PRRT rather than peptide receptor radionuclide therapy. The complete term should have been searched
3. 177 Lu-dotatate, 177 lutetium dotatate, 90 Yttrium should also have been searched
Author Response
Point 1: Databases such as PsychInfo and CINAHL should have been searched, especially for quality of life, sexual function literature that may not be included in the medical/scientific literature
Response 1: We thank the reviewer for this excellent point and as suggested we included in our literature search the APA PsycInfo database and this allowed us to include two more studies in our manuscript. Text and tables have been modified accordingly. We regret that the CINAHL database was not included in our literature search because we have not access to it.
Point 2: abbreviations appear to have been used for search terms, eg. PRRT rather than peptide receptor radionuclide therapy. The complete term should have been searched
Response 2: We thank the reviewer for this advice. As just indicated for “radioligand therapy” and “RLT” we also considered both “PRRT” and “peptide receptor radionuclide therapy” in the research term strategy. We apologize for this oversight and we have corrected the text (page 2 line 86).
Point 3: 177 Lu-dotatate, 177 lutetium dotatate, 90 Yttrium should also have been searched
Response 3: This is a valid suggestion and we added these terms in our revised literature search. Additional details on the combinations of the MeSH terms and keywords are now provided as supplemental material.
Reviewer 4 Report
Overall, an interesting summary on HRQoL in NET patients treated with PRRT put together by the authors.
A few minor comments, below.
A detailed discussion of the EORTC C-30 and GI-NET-21 forms in section 2.3 of the MS does not seem to me to be appropriate. Both EORTC forms are very widely known and used in most NET-related research and have always been treated together for very many years for NETs of both GI and pancreas origin.
That this has not been widely used in clinical trials using PRRT is a surprising finding by the authors of the analysis.
My guess is that most researchers in their reports on the efficacy of PRRT omitted the additional analysis related to quality of life assessment.
Therefore, I believe that section 2.3 does not need to be as extensively expanded and analyzed as it is in the posted MS.
Additionally, minor comments below
Discussion
Line 279 ref. Nr 34 Friend, E.; Gray, D.; Ortega, P.F.; Mcnamara, M, et al. It is already published please add the journal to the ref. It is not a congress report.
Line 281, please do not use pNEN should be panNEN or panNET, because most of EORTC QoL I mean GI-NET C-21 or the new one panNET ,consider NETG1 or NETG2 or NETG3, the clinical behavior of NEC or MINEN is different, so we can use general QoL C-30 but not those more specific NET for GI or pancreas (the no one).
Line 285 I think the authors mean GI-NET-21, which is dedicated for GI not for panNET, the C-30 is the same for both GI and panNET.
Author Response
Point 1: A detailed discussion of the EORTC C-30 and GI-NET-21 forms in section 2.3 of the MS does not seem to me to be appropriate. Both EORTC forms are very widely known and used in most NET-related research and have always been treated together for very many years for NETs of both GI and pancreas origin.
That this has not been widely used in clinical trials using PRRT is a surprising finding by the authors of the analysis.
My guess is that most researchers in their reports on the efficacy of PRRT omitted the additional analysis related to quality of life assessment.
Therefore, I believe that section 2.3 does not need to be as extensively expanded and analyzed as it is in the posted MS.
Response 1: We agree with the reviewer that it is possible that most researchers in their reports on the efficacy of PRRT omitted the additional analysis related to quality of life assessment. Of course, this explanation is speculative, therefore at this time we can draw conclusions only through the published data.
As suggested by the reviewer we revised section 2.3 to summarize the main aspects of the EORTC C-30 and GI-NET-21. More details are now provided as supplemental material.
Point 2: Line 279 ref. Nr 34 Friend, E.; Gray, D.; Ortega, P.F.; Mcnamara, M, et al. It is already published please add the journal to the ref. It is not a congress report.
Response 2: Done.
Point 3: Line 281, please do not use pNEN should be panNEN or panNET, because most of EORTC QoL I mean GI-NET C-21 or the new one panNET ,consider NETG1 or NETG2 or NETG3, the clinical behavior of NEC or MINEN is different, so we can use general QoL C-30 but not those more specific NET for GI or pancreas (the no one).
Response 3: Done.
Point 4: Line 285 I think the authors mean GI-NET-21, which is dedicated for GI not for panNET, the C-30 is the same for both GI and panNET.
Response 4: The sentence has been modified as suggested (page 12, line 297).